# Can PSMA-Targeting Radiopharmaceuticals Be Useful for Detecting Brain Metastasis of Various Tumors Using Positron Emission Tomography?

**DOI:** 10.3390/cancers17183088

**Published:** 2025-09-22

**Authors:** Esra Arslan, Nurhan Ergül, Rahime Şahin, Ediz Beyhan, Özge Erol Fenercioğlu, Yeşim Karagöz, Arzu Algün Gedik, Yakup Bozkaya, Tevfik Fikret Çermik

**Affiliations:** 1Clinic of Nuclear Medicine, University of Health and Sciences Turkey, Istanbul Training and Research Hospital, 34098 Istanbul, Turkey; 2Department of Radiology, University of Health and Sciences Turkey, Istanbul Training and Research Hospital, 34098 Istanbul, Turkey; 3Department of Pathology, University of Health and Sciences Turkey, Istanbul Training and Research Hospital, 34098 Istanbul, Turkey; 4Department of Medical Oncology, University of Health and Sciences Turkey, Istanbul Training and Research Hospital, 34098 Istanbul, Turkey; dr_yakupbozkaya@hotmail.com

**Keywords:** ^68^Ga-PSMA-11, ^18^F-FDG PET/CT, brain metastasis

## Abstract

The aim of this prospective study was to investigate the diagnostic value of ^68^Ga-PSMA-11 PET/CT by comparing ^68^Ga-PSMA-11 PET/CT, ^18^F-FDG PET/CT, and MRI findings in patients with brain metastases (BM). Given the absence of PSMA expression in normal brain parenchyma, we aimed to demonstrate the value of ^68^Ga-PSMA-11 over ^18^F-FDG in BM imaging. Twenty-four (89%) patients were included in the study for restaging, two (7%) patients for local recurrence assessment, and one (4%) patient for local recurrence and suspicion of additional lesions. The indications for ^18^F-FDG PET/CT were breast carcinoma for 37% (n:10), followed by lung carcinoma (26% (n:7), colorectal adenocarcinoma for 14% (n:4), squamous cell larynx carcinoma for 7% (n:2), gastric signet ring cell carcinoma for 4% (n:1), pancreatic neuroendocrine tumor grade 3 for 4% (n:1), thyroid papillary carcinoma for 4% (n:1), and malignant melanoma for 4% (n:1). ^68^Ga-PSMA-11 PET/CT revealed PSMA-positive brain metastases in 26 of 27 patients; only a single patient exhibited PSMA-negative lesion. This patient was followed with a diagnosis of primary larynx squamous carcinoma and had a mass suspected of metastasis. Further tests and an MRI revealed that the lesion in this patient was a hemorrhagic metastasis. In addition to its diagnostic value, we believe our findings may influence PSMA-targeted therapeutic approaches and provide a new perspective on the treatment of patients with BM.

## 1. Introduction

Brain metastasis (BM) is much more common than primary central nervous system tumors and is reported in approximately 10% to 30% of patients with metastatic disease. Although survival differs according to the histological type, grade, and molecular subtypes of the primary tumor, BM often have limited treatment response and poor prognosis [1]. BM is frequently observed in lung, breast, renal, and colorectal cancers and malignant melanoma. Treatment options generally include whole-brain radiotherapy, surgical resection, stereotactic radiosurgery, and chemotherapy [2]. 

Magnetic resonance imaging (MRI) is frequently used and the gold standard in the diagnosis and follow-up of glial tumors. However, it still has limitations for detecting viable tumor localization in previously treated cases and for differential diagnosis of necrosis from recurrence [3]. ^18^F-fluoro-2-deoxy-glucose (^18^F-FDG), positron emission tomography/computed tomography (PET/CT), which is widely used in the staging, restaging, and treatment response assessment of many cancers, has a low specificity in BM, due to high physiological radiopharmaceutical uptake. Other disadvantages such as inflammatory uptake secondary to radiation therapy and limited discrimination of low- and high-grade lesions may all cause false negative and positive evaluations [4].

Radio-labeled amino acids have been used in BM imaging for many years in oncology practice [5], and the most widely used one is 11C-methyl-L-methionine (MET), an essential amino acid labeled with the Carbon-11 isotope [6,7]. Due to the short half-life (20 minutes) of carbon-11, it only allows for PET imaging in centers with on-site cyclotrons. In their study, Galldiks et al. elaborated and presented recommendations for the use of PET imaging in the clinical management of patients with BM based on evidence from histology and/or clinically validated studies [8].

Prostate-specific membrane antigen (PSMA) is a type II transmembrane protein and has an important role, especially in high-grade prostate cancer management due to high tumor cell affinity [9]. PSMA does not distribute in the cerebrum, cerebellum, or cerebrospinal fluid. However, it can be expressed at high levels in the vascular endothelium of tumors, and its high expression associated with neovascularization, especially in high-grade gliomas, has been reported in some studies [10,11]. In addition, increased PSMA expression in tumor tissue was significantly associated with poor survival [12]. By utilizing the high tumor-related vascular expression of PSMA, it can successfully be used for the screening of distant organ metastasis not only in prostate cancer but also in different cancer types [13]. Currently, PSMA has been introduced as a promising molecular imaging target to monitor glial tumors, which are the most common primary tumors of the brain, especially in the postoperative period [14]. The aim of this study is to assess the use of ^68^Ga-PSMA-11 PET/CT in detecting BM, in comparison with ^18^F-FDG PET/CT, and its potential contribution to MRI, as well as to predict the potential of ^177^Lu-PSMA-617 treatment with these preliminary findings for the near future.

## 2. Material and Method

### 2.1. Patients

This prospective study included patients who were known to have brain metastases and were scheduled to undergo imaging for systemic screening, who were previously treated for brain metastases and suspected of local recurrence, or who had widespread metastases, and who also underwent ^18^F-FDG PET/CT imaging and subsequently underwent brain MRI due to suspicion of brain metastasis, between February 2021 and March 2022. When the study was designed, the number of participants was planned to be 25–30 and a total of 27 patients were included in the study. All patients who gave consent underwent ^68^Ga PSMA PET CT imaging.

^68^Ga-PSMA-11 PET/CT imaging was performed in 27 (11 F, 16 M) patients with a mean age of 59.48 ± 12.21 (range: 34 to 82) years. 

This prospective clinical study was approved by the local (Istanbul Training and Research Hospital) ethics committee (approval code 2021/2669). In addition, verbal and written consent was obtained from all included patients, allowing the use of their medical findings for research purposes.

### 2.2. ^18^F-FDG PET/CT Imaging

Patients with a blood glucose level of <150 mg/dl after at least 6 hours of fasting before application were included in the study. A standard activity of 3.7–5.2 MBq/kg ^18^F-FDG was intravenously injected. Approximately 60 minutes following the injection, whole-body PET/CT, including the vertex to the upper thigh area, was performed in the supine position, followed by dedicated brain PET/CT imaging. Imaging was performed with the Siemens mCT Biograph 20 ultraHD LSO PET/CT device (Siemens Molecular Imaging, Hoffmann Estates, IL, USA). For the calculation of SUVmax, regions of interest (ROI), including the highest involvement area in the BM focus, were drawn over PET sections. SUVmax was calculated according to the formula of the maximum activity in the ROI (MBq/ml)/injected ^18^F-FDG dose (MBq/kg body weight).

### 2.3. ^68^Ga-PSMA-11 PET/CT Imaging

All patients were intravenously injected with standard activity of 175 MBq (range 77–350 MBq) ^68^Ga-PSMA-11. Whole-body and -brain PET/CT imaging was performed in the supine position, including the area from the vertex to the upper part of the thigh 60 minutes after injection. ^68^Ga-PSMA-11 PET/CT imaging was performed using the same PET/CT system and procedures as the ^18^F-FDG PET/CT imaging. For the calculation of SUVmax, regions of interest (ROI), including the highest involvement area in the brain metastasis focus, were drawn over PET sections.

The evaluation of PET/CT imaging with both radiopharmaceuticals was performed visually as well as semi-quantitatively using SUVmax of BM lesions calculated for all patients. The obtained PET parameters were comparatively analyzed and correlated with the clinical characteristics of the patients, such as demographics, BM primary focus, and histopathological data (Table 1).

### 2.4. IHC Staining with PSMA

Immunohistochemistry (IHC) was performed using the prostate-specific membrane antigen (PSMA) (EP192, Epitomics, Inc. Burlingame, CA, USA) Rabbit Monoclonal Primary Antibody (Roche/Cell Marque). The antibody was applied on formalin-fixed, paraffin-embedded (FFPE) tissue sections prepared from both primary tumor lesions and metastatic tumor tissue. Paraffin blocks were cut into 2 µm sections, deparaffinized, and rehydrated through graded alcohols followed by washing in Tris-buffered saline (TBS, Aniara Diagnostica LLC/Medicago AB, Sweden). Tissue sections were subsequently stained on the Ventana Benchmark IHC/ISH automated platform. Slides were incubated for 20 minutes at room temperature with the anti-PSMA rabbit monoclonal antibody (clone EP192) at a dilution of 1:20, and visualization was achieved using the ultraView Universal DAB Detection Kit (Ventana Medical Systems, Roche, Switzerland).

#### Scoring System

PSMA expression was assessed by two parameters: staining percentage (SP) and staining intensity (SI).

SP was visually quantified using a four-tiered system:
0: SP <20% (no expression).1: SP 20–50% (low expression).2: SP 50–80% (moderate expression).3: SP >80% (high expression).
SI was also evaluated on a four-tiered scale:
0: No staining.1: Weak.2: Moderate.3: Strong.
The final IHC score was calculated by multiplying the staining percentage score by the staining intensity score. Both membranous and cytoplasmic PSMA expression were recorded (Table 2).

### 2.5. Statistical Analysis

Version 21.0 of the SPSS (Statistical Package for the Social Sciences) program (IBM, Armonk, NY, USA) was used for statistical analysis. Descriptive statistics were expressed as mean ± standard deviation or median (minimum–maximum) for discrete and continuous numerical variables and the number of cases and (%) for categorical variables.

## 3. Results

Twenty-four (89%) patients were included in the study for restaging, two (7%) patients for local recurrence assessment, and one (4%) patient for local recurrence and suspicion of additional lesions. The indications for ^18^F-FDG PET/CT were breast carcinoma (Figure 1) for 37% (n:10), followed by lung carcinoma for 26% (n:7), colorectal adenocarcinoma for 14% (n:4), squamous cell larynx carcinoma for 7% (n:2), gastric signet ring cell carcinoma for 4% (n:1), pancreatic NET3 (Figure 2) for 4% (n:1), papillary thyroid carcinoma for 4% (n:1), and malignant melanoma for 4% (n:1). ^68^Ga-PSMA-11 PET/CT revealed PSMA-positive brain metastases in 26 of 27 patients; only a single patient exhibited PSMA-negative lesion (Figure 3). This single patient was followed with a diagnosis of primary larynx squamous carcinoma and had a mass suspected of metastasis. Further tests and an MRI revealed that the lesion in this patient was a hemorrhagic metastasis.

The histopathological diagnosis was made by biopsy material obtained on the primary tumor (Table 1). Cranial MRI was performed in all cases within the same week with ^68^Ga-PSMA-11 PET/CT, and all metastatic lesions were confirmed with MRI. MRI revealed a single metastatic lesion in 15 (55%) patients and multiple metastatic lesions in 12 (45%) patients. Immunohistochemical staining with PSMA was performed in primary tumors of 10 patients, and positive staining was detected in 5 (50%) of them, as well as in all 4 (100%) brain metastatic tumor foci (Table 2).
cancers-17-03088-t001_Table 1Table 1Clinical characteristics of patients *.NoAgeGenderPrimary Histopathological DiagnosisNumber of Metastatic LesionsAxial Diameter of the Most Prominent Metastatic Lesion (cm)^18^F- FDG SUV Max of the Most Prominent Metastatic Lesion^68^Ga-PSMA-11 SUVmax of the Most Prominent Metastatic LesionSmallest Axial Diameter of the Lesion Detected with ^68^Ga-PSMA-11 (cm)Lesions Observed in ^68^Ga-PSMA-11 PET/CT not Observed in ^18^F- FDG PET/CT**1**47FemaleTriple-negative ID Breast Carcinoma14.3411.976.58Not applicableNot applicable**2**51FemaleTriple-negative ID Breast CarcinomaMultiple1.1323.985.550.71Not applicable**3**53FemaleHER2 Positive ID Breast CarcinomaMultiple2.4115.6913.830.40Not applicable**4**65FemaleLuminal B ID Breast Carcinoma32.3017.417.72Not applicableNot applicable**5**34FemaleLuminal B ID Breast Carcinoma12.7720.1910.77Not applicableNot applicable**6**41FemaleHER2 Positive ID Breast Carcinoma11.2021.7914.50Not applicableNot applicable**7**43FemaleLuminal B ID Breast CarcinomaMultiple2.0830.674.490.22**+****8**47FemaleLuminal B ID Breast CarcinomaMultiple0.57No uptake2.110.26**+****9**61MaleSquamous Cell Lung CarcinomaMultiple1.4511.804.380.46**+****10**56FemaleLung Adenocarcinoma12.34No uptake2.190.39**+****11**71MaleSquamous Cell Lung Carcinoma13.1910.554.35Not applicableNot applicable**12**67MaleSmall Cell Lung CarcinomaMultiple1.279.333.590.31Not applicable**13**72MaleColorectal Adenocarcinoma12.005.603.80Not applicableNot applicable**14**51MaleColorectal Adenocarcinoma11.60No uptake2.801.60**+****15**67MaleColorectal AdenocarcinomaMultiple2.4611.726.510.35**+****16**74MaleColorectal Adenocarcinoma41.3934.878.290.54Not applicable**17**68MalePancreatic Neuroendocrine Tumor Grade 3Multiple3.0220.2912.441.32**+****18**47MaleSignet Ring Cell Gastric Carcinoma32.2832.003.77Not applicableNot applicable**19**64MaleFollicular Variant of Papillary Thyroid Carcinoma11.16No uptake24.951.16**+****20**63MaleSquamous Cell Lung Carcinoma11.7629.376.891.76Not applicable**21**56MaleLung Adenocarcinoma12.81No uptake2.72.81**+****22**60MaleSquamous Cell Lung Carcinoma + Squamous Cell Nasopharnygeal Carcinoma13.7312.272.23Not applicableNot applicable**23**51FemaleTriple-negative ID Breast Carcinoma21.69.554.30.76**+****24**81FemaleLuminal B ID Breast Carcinoma12.0519.684.18Not applicableNot applicable**25**71MaleMalignant MelanomaMultiple3.239.727.080.48**+****26**63MaleSquamous Cell Larynx Carcinoma + Squamous Cell Lung Carcinoma12.18No uptake2.92.18**+****27**82MaleSquamous Cell Larynx Carcinoma11.7No uptakeNo uptakeNot applicableNot applicable*** ID:** Invasive Ductal.
cancers-17-03088-t002_Table 2Table 2Immunohistochemical staining with PSMA in primary tumors and metastatic foci *.NPrimary Histopathological DiagnosisPrimary TumorPSMA Staining ScoreMetastatic TumorPSMA Staining Score1.Triple-negative ID Breast CarcinomaSP:11 × 1:1SP:11 × 1:1SI:1SI:12.Triple-negative ID Breast CarcinomaSP:00SP:00SI:0
SI:0
13.Colorectal AdenocarcinomaSP:11 × 1:1SP:11 × 1:1SI:1
SI:116.HER2 Positive ID Breast CarcinomaSP:11 × 1:1SP:22 × 1:2SI:1
SI:1
18.Luminal B ID Breast CarcinomaSP:11 × 1:1SP:00SI:1
SI:0
10.Lung AdenocarcinomaSP:00SP:00SI:0
SI:0
12.Small Cell Lung CarcinomaSP:00SP:00SI:0
SI:0
3.HER2 Positive ID Breast CarcinomaSP:00SP:00SI:0
SI:0
5.Luminal B ID Breast CarcinomaSP:0
SP:00SI:00SI:0
8.Luminal B ID Breast CarcinomaSP:11 × 1:1SP:22 × 1:2SI:1
SI:1
*** ID:** Invasive Ductal; **SP**: Staining Percentage; **SI:** Staining Index.


While BM foci were detected in all 26 patients with ^68^Ga-PSMA-11 PET/CT, only 15 (58%) of them could be detected with ^18^F-FDG PET/CT. The mean ± SD axial diameter of the largest lesions visualized on ^68^Ga-PSMA-11 PET/CT was found to be 2.16 ± 0.88 cm (range: 0.57–4.34 cm). The smallest size of metastases that could be visualized was 0.22 cm in axial diameter (Figure 1). Mean ± SD SUVmax of metastatic lesions measured on ^18^F-FDG PET/CT imaging was 17.9 ± 8.6 (range: 5.6–37.8), and mean ± SD SUVmax measured on ^68^Ga-PSMA-11 PET/CT imaging was 6.8 ± 5.2 (range: 2.1–24.9). Although the mean SUVmax was higher in ^18^F-FDG than ^68^Ga-PSMA-11, no statistically significant difference was found between the two imaging methods (*p*:0.37). 

## 4. Discussion

The tumor microenvironment plays a crucial role in the prognostic behavior of malignancies. Angiogenesis is one of the most critical pillars of tumor progression, which starts with the stimulation of endothelial cells mediated mainly by vascular endothelial growth factor (VEGF) due to hypoxia and molecular changes [15]. Wernicke et al. reported that PSMA expression was detected in tumor endothelium in all 32 patients with glioblastoma multiforme. Intense PSMA staining was reported in 22 cases (69%) [16]. Tanjore et al. demonstrated the association of neovascular PSMA expression with malignant progression and poor survival after a 10.4-year follow-up of 371 patients with glioblastoma and 52 patients with brain metastatic lung cancer [17]. One of the most common problems in the clinical management of primary brain tumors is the accurate detection of local recurrence during postoperative follow-up. On the other hand, Akgun et al. reported significantly higher PSMA expression in grade IV glial tumors than grade II and III with ^68^Ga-PSMA-11 PET/CT imaging [14].

In metastatic brain tumors, there is a very heterogeneous distribution for the primary tumor type such that in 15% of patients with BM diagnosis, primary malignant tumor cannot even be detected. Lung cancer, breast cancer, and malignant melanoma have frequently been documented as primaries for patients with BM. In their study, Habbous et al. screened patients with a metastatic brain tumor diagnosis between 2010 and 2018, and reported that BM was found in 25.478 (4.2%) of the 601.678 patients, where, the primary focus was lung cancer in 60%, breast cancer in 11%, and malignant melanoma in 6% [18]. BM is seen throughout the disease in approximately 40–50% of patients with lung cancer, especially small cell lung cancer, and this rate is around 10–20% at the time of initial diagnosis [2]. In breast cancers, approximately 15–20% BM is reported. HER-2 positive breast cancer especially has a high BM rate [19]. In our study, the most common BM focus was breast cancer with a rate of 38%, which id consistent with the published data, followed by lung cancer at 27% and colon cancer at 15%. The reason why our rate of lung cancer cases is lower than the literature may be related to the fact that our hospital is a reference center especially for breast and colorectal cancers. Wernicke et al. reported BM in 14 of 106 patients with breast cancer. In the immunohistochemical staining to investigate the PSMA expression in tumor vascular endothelial cells, positive PSMA expression was observed in 74% (68/92) of primary breast cancers and 100% (n = 14) in BM. With this study, the researchers demonstrated the high expression of PSMA associated with tumor neovascularization [20]. In the study, Nomura et al. investigated the tissues targeted by PSMA immunohistochemically on 19 patients with glioma and 5 with breast cancer, all with a secondary BM diagnosis. At the same time, no PSMA staining was detected in normal brain microvasculature; a significantly increased (*p* < 0.05) rate of PSMA expression was determined in high angiogenic grade IV glial tumors. In addition, more vital vascular PSMA staining was detected in BM compared to primary breast cancer tumor tissue. Researchers have demonstrated the high expression of PSMA in primary tumor and BM-associated vascular tissues [21].

BM-associated PSMA expression has also been coincidentally documented for different cancer types in the literature. Coincidentally, Hod et al., in their 76-year-old male patient with resected malignant melanoma, observed unexpectedly high PSMA brain involvement in the previous surgical cavity on ^68^Ga-PSMA-11 PET/CT imaging. After CT and MRI, the diagnosis of local melanoma recurrence was confirmed [22]. Vallejo-Armenta et al. reported significantly increased uptake in metastatic brain tumors, high-grade glial brain tumors, and recurrent gliomas with preoperative [^99m^Tc] Tc-iPSMA SPECT brain imaging in 41 patients diagnosed with BM by MRI. However, they did not detect involvement in low-grade glial brain tumor lesions. Researchers reported that PSMA is expressed at high levels in grade IV glioma and BM vascular endothelium [23]. In support of all these studies, tumor-associated neovascularization of a 47-year-old patient with triple-negative breast carcinoma BM with low ^18^F-FDG uptake despite high PSMA activity was demonstrated [24]. In the present study, all BM foci that could not be detected with ^18^F-FDG PET/CT were successfully detected with ^68^Ga-PSMA-11 PET/CT in 42% of the cases. Wei et al. observed significant tumor regression and decreased PSMA expression in cerebral lesions based on ^68^Ga-PSMA PET/CT imaging performed after ^177^Lu-PSMA-617 treatments combined with radiotherapy in two prostate cancer patients with cerebral metastasis [25]. Dall’Armellina S et al. also presented results of PSMA-targeted PET imaging for brain metastases from non-prostate solid tumors, including results from a total of 23 studies [26].

However, false positives associated with ^68^Ga-PSMA-11 PET/CT imaging should not be ignored. Noto et al. observed focal involvement in the right frontal lobe with suspected BM in a 65-year-old patient diagnosed with prostate adenocarcinoma in ^68^Ga-PSMA-HBED-CC-PET/CT imaging. Eventually, it was understood that the reason for the cerebral ^68^Ga-PSMA involvement observed in that patient was recent cerebral infarction [27]. 

Oh G et al. reported a case of subacute cerebellar infarction mimicking metastasis on prostate-specific membrane antigen (PSMA) PET/CT in a 77-year-old man with prostate cancer [28]. And Jain V et al. also published a case report in which they observed positive uptake on 18 F-PSMA PET/CT due to subacute cerebral infarction [29]. Additionally, Huang Y. et al. observed false-positive findings on a ^68^Ga-labeled prostate-specific membrane antigen (PSMA) ligand PET/CT scan due to elevated prostate-specific antigen levels in a 56-year-old man with a history of brain abscess [30].

Currently, PSMA-targeted molecular imaging using different radiopharmaceuticals is established mostly for prostate cancer, owing to the higher expression of PSMA in especially aggressive prostate cancer cells. The introduction of theragnostic applications with the use of the PSMA target has recently provided a paradigm change for the metastatic castration-resistant prostate cancer and studies are on the way to assess its use for the earlier periods of the disease. Our study along with aforementioned studies and some others in the literature definitely support the use of the PSMA target for tumors other than prostate cancer, thanks to its overexpression related to neovascularization regardless of the tumor type to the best of our knowledge and, in particular, to detect BM owing to its cerebral biodistribution advantage for healthy brain tissue over the most commonly used oncologic PET radiopharmaceutical ^18^F-FDG. In this regard, PSMA seems to be a promising target for patients with various tumors with extensive metastasis including BM to provide an alternative systemic treatment through theragnostic applications in the near future.

## 5. Conclusions

This study demonstrates that the absence of PSMA expression in normal cerebral and cerebellar parenchyma, in contrast to the high physiological background uptake of ^18^F-FDG, provides ^68^Ga-PSMA-11 PET/CT with a clear diagnostic advantage in BM imaging. Although PSMA is frequently upregulated in high-grade tumors, this cannot be generalized across all malignancies, underscoring the need for larger studies that systematically correlate PSMA immunohistochemistry (IHC) scores with ^68^Ga-PSMA-11 uptake across tumor types and histopathological subgroups. Importantly, treatment-related factors such as prior radiotherapy, systemic chemotherapy, and hormone therapy, which may alter receptor expression, should also be addressed in future analyses.

Our findings further highlight the potential of PSMA as a therapeutic target in BM, supporting the development of radionuclide-based strategies capable of exploiting tumor neovascularization while preserving normal brain tissue. To validate this approach and determine its true impact on patient outcomes, well-designed prospective trials incorporating standardized brain MRI alongside ^18^F-FDG PET/CT and ^68^Ga-PSMA-11 are warranted. Ultimately, this strategy may contribute to the establishment of innovative therapeutic options for patients with BM from diverse primary tumors.

## Figures and Tables

**Figure 1 cancers-17-03088-f001:**
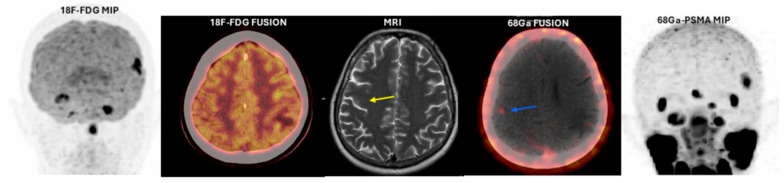
(**Patient No: 7**)**:** A 43-year-old woman with ID breast carcinoma; Estrogen Receptor 95%, Progesterone Receptor 95%, c-erbB2 (–), Ki67: 45%. Metastatic lesion with an axial diameter of 0.22 cm detected by ^68^Ga-PSMA-11 PET/CT and MRI (T2-weighted). *PET: positron emission tomography; MIP: maximum intensity projection; ^18^F-FDG: ^18^F-fluorodeoxyglucose; ^68^Ga PSMA: Gallium-68-prostate-specific membrane antigen (lesion detected with MRI: yellow arrow, ^68^Ga PSMA-11 fusion image positive lesion: blue arrow)*.

**Figure 2 cancers-17-03088-f002:**
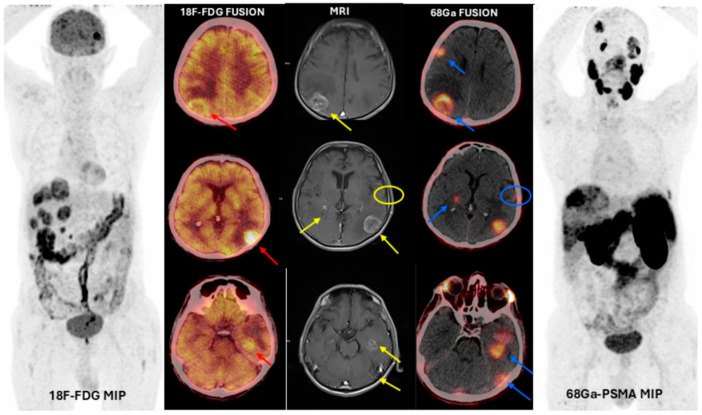
(**Patient No: 17**)**:** A 68-year-old man, brain metastases of primary pancreatic neuroendocrine tumor grade 3 detected with ^68^Ga-PSMA-11 PET/CT and MRI (T1-weighted)*. **PET:** positron emission tomography; **MIP:** maximum intensity projection; **^18^F-FDG**: ^18^F-fluorodeoxyglucose; **^68^Ga PSMA:** Gallium-68-prostate-specific membrane antigen (^18^F- FDG fusion images positive metastatic foci: red arrows, lesions detected with MRI: yellow arrows and yellow circle, ^68^Ga PSMA-11 fusion images, positive metastatic foci: blue arrows and blue circle)*.

**Figure 3 cancers-17-03088-f003:**
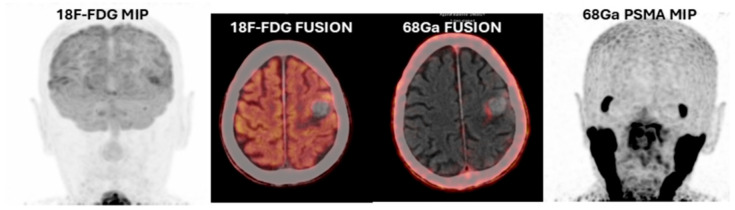
(**Patient No: 27**)**:** An 82-year-old man, squamous cell larynx carcinoma hemorrhagic metastasis (*^18^F- FDG negative fusion and ^68^Ga PSMA-11 negative fusion images and MRI)*.

## Data Availability

The data presented in this study are available in this article.

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
