# Peer review of "Can PSMA-Targeting Radiopharmaceuticals Be Useful for Detecting Brain Metastasis of Various Tumors Using Positron Emission Tomography?"

_cancers, 2025, doi:10.3390/cancers17183088_

Round 1

Reviewer 1 Report

Comments and Suggestions for Authors

This is an interesting paper that adds a potential additional indication for PSMA PET/CT imaging.  The authors found additional PSMA-detected brain metastases that were not detectable on FDG- PET/CT imaging.  This opens potential diagnostic and therapeutic targets for patients with brain metastases.

However, there are several areas that should be qualified in the study.  First- in terms of methods, it is unclear how the patients were selected.  I see that this was a prospective clinical trial, but there is no mention of expected recruitment numbers.  Did all patients who consented receive PSMA PET imaging? All of the patients included had PSMA positive brain metastases - and it would be important to know if all patients meeting these standards or only some of the patients meeting these standards were included. 

The current gold standard for detection of brain metastases is a contrast MRI of the brain. While all patients had brain MRI scans confirming metastatic lesions (this should be noted in the materials and methods section, as it is part of the protocol), there is no mention if all metastatic lesions were visible on PSMA PET imaging.  This is crucially important - if there are lesions not detected by PSMA, it potentially limits the ability of this imaging as a diagnostic and therapeutic tool. If all lesions on MRI were not detected by PSMA PET/CT imaging, please discuss this in the text, and provide further data regarding this detail (add to table 1, or provide clarifying details in text). 

Author Response

Reviewer 1

Comments and Suggestions for Authors

This is an interesting paper that adds a potential additional indication for PSMA PET/CT imaging.  The authors found additional PSMA-detected brain metastases that were not detectable on FDG- PET/CT imaging.  This opens potential diagnostic and therapeutic targets for patients with brain metastases.

However, there are several areas that should be qualified in the study.

Q1.First- in terms of methods, it is unclear how the patients were selected. 

A1.This prospective study included patients who were known to have brain metastases and were scheduled to undergo imaging for systemic screening, who were previously treated for brain metastases and suspected of local recurrence, or who had widespread metastases, and who also underwent 18F-FDG PET/CT imaging and subsequently underwent brain MRI imaging due to suspicion of brain metastasis, between February 2021 and March 2022.

Added (Page3 Line 119-124)

Q2.I see that this was a prospective clinical trial, but there is no mention of expected recruitment numbers.

A2.When the study was designed, the number of participants was planned to be 25-30 and a total of 27 patients were included in the study.

Added (Page3 Line 124-125)

Q3.Did all patients who consented receive PSMA PET imaging?

A3.All patients who gave consent underwent 68Ga PSMA PET CT imaging.

Added Page3 Line 125-126)

Q4.All of the patients included had PSMA positive brain metastases - and it would be important to know if all patients meeting these standards or only some of the patients meeting these standards were included. 

A4.26/27 included patients had PSMA-positive brain metastases with 68 Ga PSMA PET CT but only one patient had PSMA-negative brain metastases. This single patient was followed with a diagnosis of primary laryngeal squamous carcinoma and had a mass suspected of metastasis. Further tests and an MRI revealed that the lesion in this patient was a hemorrhagic metastasis.

(Added Page 4 Line 185-189)

Q5.The current gold standard for detection of brain metastases is a contrast MRI of the brain.

While all patients had brain MRI scans confirming metastatic lesions (this should be noted in the materials and methods section, as it is part of the protocol), there is no mention if all metastatic lesions were visible on PSMA PET imaging. 

A5.Mentioned as adviced  added Page 3 Line120-125

Q6.This is crucially important - if there are lesions not detected by PSMA, it potentially limits the ability of this imaging as a diagnostic and therapeutic tool.

A6.There was only one brain metastatic lesion not detected by PSMA and was added to Table 1 at your suggestion. This case was interpreted as hemorrhagic metastasis from larynx carcinoma in the MRI report. Radiopharmaceutical uptake was not observed on either 18F-FDG PET/CT or 68Ga PSMA PET/CT (Fig. ????).Added(Page 4 Line186-189 and Table 1)

Q7.If all lesions on MRI were not detected by PSMA PET/CT imaging, please discuss this in the text, and provide further data regarding this detail (add to table 1, or provide clarifying details in text). 

A7.At your suggestion, the necessary additions and changes were made to both the text and Table 1.(Page 6 Table 1)

Reviewer 2 Report

Comments and Suggestions for Authors

Arslan et al proposed the use of prostate specific membrane antigen (PSMA) as a PET target for non-prostatic brain metastases, with hypothetically improved detection due to the lack of presence of background PSMA in normal brain parenchyma.  Patients with brain metastases from a variety of primary sites (breast, lung, colorectal, laryngeal, gastric, pancreatic neuroendocrine, thyroid papillary and melanoma) who underwent MRI and 18 FDG PET in the course of their evaluation underwent 68Ga-PSMA-11 PET.  Findings were that the PSMA PET detected more lesions than FDG PET.

Simple summary: The simple summary did not contain the actual results, describing the hypothesis and then indicating that PSMA PET had diagnostic value but did not document the actual results.  There  should be a sentence detailing these results.

Abstract: The abstract provided an adequate summary of the submission. Note that lines 47-48 state “68Ga-PSMA-11 imaging is concluded to be superior to FDG imaging, for detection of BM in various tumors. ”  The superiority of PSMA PET cannot be concluded from this study, and alternative wording would be “results suggest that 68Ga-PSMA-11 PET imaging is superior to FDG PET in detecting brain metastases pending further studies. ”  This is further discussed in the discussion and conclusion sections below.

Introduction:  The introduction provided a background to the use of imaging techniques for brain metastases, including MRI and 18 FDG PET, and the rationale for using PSMA PET.  

Materials and methods:  Straightforward, describing the patient inclusion criteria, radiographic methods, pathology and statistical methodology.

Results:  Provides an adequate summary of the results.  Figures 1-6 are excessive (see comments on figures and tables below)

Discussion:  The authors provided an understandable discussion of their findings, including previously published data supporting their findings, and prospects for future development.  There is also a typographical error on line 297: “use od” should read as “use of.”

Conclusions:  The conclusion (and the discussion) supported the superiority of PSMA targeted molecular imaging over FDG PET for the evaluation of non-prostatic brain metastases.  However, in the conclusion section (lines 309-310) it was stated that the study " clearly demonstrated the superiority of 68Ga-PSMA-11 over FDG PET/CT for BM imaging." However, the study shows to my reading that while supported as possibly superior, further studies (providing standardized imaging with brain MRI, FDG PET in PSMA PET prospectively to all patients) are needed to determine whether it actually improves detection and also improves outcomes that would demonstrate true superiority.

References: All references are pertinent to the submission.

Figures and tables:  Figures 1-6 are excessive.  These figures show findings in individual patients with regard to findings on FDG PET versus PSMA PET.  No more than 2 figures should be sufficient to demonstrate an improved sensitivity to PSMA PET.  All tables are complementary or supplementar to the narrative text in did not represent any redundancy.

Author Response

Reviewer 2

Comments and Suggestions for Authors

Arslan et al proposed the use of prostate specific membrane antigen (PSMA) as a PET target for non-prostatic brain metastases, with hypothetically improved detection due to the lack of presence of background PSMA in normal brain parenchyma.  Patients with brain metastases from a variety of primary sites (breast, lung, colorectal, laryngeal, gastric, pancreatic neuroendocrine, thyroid papillary and melanoma) who underwent MRI and 18 FDG PET in the course of their evaluation underwent 68Ga-PSMA-11 PET.  Findings were that the PSMA PET detected more lesions than FDG PET.

Q1.Simple summary: The simple summary did not contain the actual results, describing the hypothesis and then indicating that PSMA PET had diagnostic value but did not document the actual results.  There  should be a sentence detailing these results.

A1.As recommended, a more descriptive sentence was added to the simple summary section.

Page 1 Line20-31

Abstract: The abstract provided an adequate summary of the submission. Note that lines 47-48 state “68Ga-PSMA-11 imaging is concluded to be superior to FDG imaging, for detection of BM in various tumors. ”  The superiority of PSMA PET cannot be concluded from this study, and alternative wording would be “results suggest that 68Ga-PSMA-11 PET imaging is superior to FDG PET in detecting brain metastases pending further studies.

Q2.”  This is further discussed in the discussion and conclusion sections below.

A2.Following your suggestion, the comment was rewritten to avoid definitive statements and additions were made.Page   Line  65-74

Introduction:  The introduction provided a background to the use of imaging techniques for brain metastases, including MRI and 18 FDG PET, and the rationale for using PSMA PET.

Materials and methods:  Straightforward, describing the patient inclusion criteria, radiographic methods, pathology and statistical methodology.

Q3. Results:  Provides an adequate summary of the results.  Figures 1-6 are excessive (see comments on figures and tables below)

A3.The number of figures has been reduced and simplified, and a case with brain metastasis observed as PSMA negative has been added, at your suggestion.

Q4.Discussion:  The authors provided an understandable discussion of their findings, including previously published data supporting their findings, and prospects for future development.  There is also a typographical error on line 297: “use od” should read as “use of.”

A4.Corrected typo as per your suggestion

Page 11 Line 295

Q5.Conclusions:  The conclusion (and the discussion) supported the superiority of PSMA targeted molecular imaging over FDG PET for the evaluation of non-prostatic brain metastases.  However, in the conclusion section (lines 309-310) it was stated that the study " clearly demonstrated the superiority of 68Ga-PSMA-11 over FDG PET/CT for BM imaging."

However, the study shows to my reading that while supported as possibly superior, further studies (providing standardized imaging with brain MRI, FDG PET in PSMA PET prospectively to all patients) are needed to determine whether it actually improves detection and also improves outcomes that would demonstrate true superiority.

A5.Added with your valuable advices to conclusion line 314-323

References: All references are pertinent to the submission.

Q6.Figures and tables:  Figures 1-6 are excessive.  These figures show findings in individual patients with regard to findings on FDG PET versus PSMA PET.  No more than 2 figures should be sufficient to demonstrate an improved sensitivity to PSMA PET.  All tables are complementary or supplementar to the narrative text in did not represent any redundancy.

A6.Upon your recommendation, the number of figures was reduced and a new non-PSMA metastatic case was added to simplify the figures.

Reviewer 3 Report

Comments and Suggestions for Authors

This is a noteworthy paper on usefulness of 68Ga-PSMA-11 PET/CT in detecting brain metastasis (BM), along with comparison with 18F-FDG-PET/CT. It showed the superiority of 68Ga-PSMA-11 PET/CT to 18F-FDG-PET/CT in detecting BM. I have the following comments about it.

#It reported that by IHC, the primary tumor was positive for PSMA in 5/10 (50%) cases, and the surgically resected metastatic tumor was positive in 4/4 (100%).

It is true that PSMA is taken up by high grade tumors with a high percentage, but not by ALL, I guess. This means that 68Ga-PSMA-11 PET/CT could fail to detect BM, right? How can we compromise with this? I believe ‘4/4 (100%)’ does not guarantee the perfect detection of BM. PROS and CONS must be there about the use of 68Ga-PSMA-11 PET/CT. Reding the Discussion, just PROS seem to be addressed, but CONS or pitfalls should also be faithfully discussed.

#This kind of overlaps with the above. In this paper, 68Ga-PSMA-11 PET/CT detected BM that could not been found by 18F-FDG-PET/CT. Seeing Table 1, the tumors detected by 68Ga-PSMA-11 PET/CT but not by 18F-FDG-PET/CT span across various tumors. But I just wonder the detection of BM by 68Ga-PSMA-11 PET/CT is influenced by the kinds (histology) of primary tumors? Are there any stories where this tumor, but not that one, is preferentially detected by 68Ga-PSMA-11 PET/CT??? This information/data is important to implement widely 68Ga-PSMA-11 PET/CT in the clinical settings. The tumors prone to brain metastasis, like melanoma or lung carcinoma, could be perfectly detected by 68Ga-PSMA-11 PET/CT?

#Table 1

-The term ‘Cancer’ is not recommended. This is because it entails many kinds of malignancy, including carcinoma or sarcoma. Not cancer but carcinoma, sarcoma, or something else, which are more pinpointed, could be advisable. This is particularly because it says ‘The histopathological diagnosis was made by biopsy material obtained on the primary tumor (Table 1)’ (page 4, line 152-153).

-Just ‘Squamous’, ‘Neuroendocrine’, or ‘Signet Ring cell’ means nothing. ‘Squamous cell carcinoma’, ‘Neuroendocrine carcinoma’ or ‘Signet Ring cell carcinoma’ should be the case.

-This concept is also true for other parts of the whole of the MS. ‘Cancer’ is a convenient word, but could be an incorrect, should-be-discouraged’ term. Please limit its use to where it could safely be applied; otherwise, it should be discouraged.  

-Some blanks are noted, but to leave blanks here is not recommended. ‘Not applicable’ or ‘-’ should be the case here.

-This is pertinent to just above. I cannot understand why No 4 and No18 are blanked in the 2nd right column although they have 3 BM foci.

#Table 2

-This is similar to the above about Table 1. The ‘Primary Diagnosis’ or ‘Histopathological Subtype’ should be re-considered. For example, why ‘Lung Cancer’ is ‘Primary Diagnosis’ and ‘Adenocarcinoma’ is ‘Histopathological Subtype’? Probably, ‘Lung Carcinoma’ is ‘Primary Diagnosis’, then comes ‘Adenocarcinoma’ as the ‘Histopathological Subtype’. Or, just ‘Lung adenocarcinoma’ would suffice as ‘Primary Diagnosis’. Anyway, this table should be accordingly revised by the help of pathologists.

-If this table intends to show ‘Staining Score’, the materials and methos should sufficiently address this, which I will tell later.

#Figure 7 and 8

The IHC photos should be shown with much higher magnification (x200 or more). With such a low mag., no one can recognize the brown pigment on endothelium within the tumor tissues.

#page 3, line 132-133

-anti-PSMA antibody: The lot number, the product ID, and/or whether they are polyclonal or monoclonal should be clearly shown. The full information of this antibody is a must.

-treated with for 1 hour and yet incubated for 20 minutes??? After all, this means that ‘incubated with the antibody for 1 hour. 20 min.’???

-Nex >>> Next

-The quantification strategy is quite unclear. What is staining score??? Why can staining score of 3 or less be low, and that of higher than 3 be high???

-After all, this section ‘IHC staining with PSMA antigen’ is extremely poorly described.

#Figure 1-8

I am afraid ‘red arrows’, ‘dashed red arrows’, or so are not colored rightly, or they do not indicate the right lesion. For example, ‘blue arrows’ in Figure 1 look red. Is this due to the conversion of digital files? Anyway, please check the thoroughness of the legends all over.

#page 4, line 151, ‘NET’: What is this NET? Is this grade 1, 2, or 3? This should be NET grade 3, I guess. A more correct, pinpointed term should be applied.

#Some minor wording/grammatical errors I have noticed. Please check the MS again.

-page 2, line 67-68: (18F-FDG), positron…(PET/CT) which is >>> (18F-FDG) positron…(PET/CT), which is

-page 3, line 121: F-18 FDG >>> 18F-FDG (?)

-Figure legend of figure 2: 50%.  .2cm axial diameter >>> 50%. 2cm axial diameter

-etc., etc., etc.

Author Response

Comments and Suggestions for Authors

This is a noteworthy paper on usefulness of 68Ga-PSMA-11 PET/CT in detecting brain metastasis (BM), along with comparison with 18F-FDG-PET/CT. It showed the superiority of 68Ga-PSMA-11 PET/CT to 18F-FDG-PET/CT in detecting BM. I have the following comments about it.

Q1.#It reported that by IHC, the primary tumor was positive for PSMA in 5/10 (50%) cases, and the surgically resected metastatic tumor was positive in 4/4 (100%).

Q2. It is true that PSMA is taken up by high grade tumors with a high percentage, but not by ALL, I guess. This means that 68Ga-PSMA-11 PET/CT could fail to detect BM, right? How can we compromise with this?

A2.  ‘’It is a known finding that PSMA is highly upregulated in high-grade tumors, but it would be inappropriate to generalize this to all tumors. Therefore, studies are needed to demonstrate the results of large series comparing tumors and their histopathological subtypes using IHC scores.’’ Added to discussion part with your advise Line 308-322

Q3.  I believe ‘4/4 (100%)’ does not guarantee the perfect detection of BM. PROS and CONS must be there about the use of 68Ga-PSMA-11 PET/CT. Reding the Discussion, just PROS seem to be addressed, but CONS or pitfalls should also be faithfully discussed.

A3. Furthermore, considering any limitations that may have impacted receptor status, such as prior RT, systemic chemotherapy, and hormonotherapy, more comprehensive and detailed analyses are clearly needed. Added to discussion part with your advice Line 306-312

#This kind of overlaps with the above. In this paper, 68Ga-PSMA-11 PET/CT detected BM that could not been found by 18F-FDG-PET/CT.

Q4.Seeing Table 1, the tumors detected by 68Ga-PSMA-11 PET/CT but not by 18F-FDG-PET/CT span across various tumors. But I just wonder the detection of BM by 68Ga-PSMA-11 PET/CT is influenced by the kinds (histology) of primary tumors? Are there any stories where this tumor, but not that one, is preferentially detected by 68Ga-PSMA-11 PET/CT??? This information/data is important to implement widely 68Ga-PSMA-11 PET/CT in the clinical settings. The tumors prone to brain metastasis, like melanoma or lung carcinoma, could be perfectly detected by 68Ga-PSMA-11 PET/CT?

A4. As I've added in the discussion section, the answer to this question can only be possible by knowing the number of patients in large series of tumor subtypes and the treatments they received that may affect their receptor levels. We hope that future studies will shed light.

#Table 1

Q5.-The term ‘Cancer’ is not recommended. This is because it entails many kinds of malignancy, including carcinoma or sarcoma. Not cancer but carcinoma, sarcoma, or something else, which are more pinpointed, could be advisable. This is particularly because it says ‘The histopathological diagnosis was made by biopsy material obtained on the primary tumor (Table 1)’ (page 4, line 152-153).

A5.Table 1, Table 2, and all histopathologically incorrectly expressed definitions in the article were tried to be corrected upon your suggestion. I am grateful for your advice.

Q6.-Just ‘Squamous’, ‘Neuroendocrine’, or ‘Signet Ring cell’ means nothing. ‘Squamous cell carcinoma’, ‘Neuroendocrine carcinoma’ or ‘Signet Ring cell carcinoma’ should be the case.

A6.All histopathologically incorrectly expressed definitions in the article were tried to be corrected upon your suggestion. Thanks again

Q7.-This concept is also true for other parts of the whole of the MS. ‘Cancer’ is a convenient word, but could be an incorrect, should-be-discouraged’ term. Please limit its use to where it could safely be applied; otherwise, it should be discouraged.  

A7.Attempts were made to correctly diagnose it as a carcinoma or neuroendocrine tumor upon your recommendation.

Q8-Some blanks are noted, but to leave blanks here is not recommended. ‘Not applicable’ or ‘-’ should be the case here.

A8.The blanks in Table 1 have been colored yellow and replaced with the term "not applicable" upon your recommendation.

Q9-This is pertinent to just above. I cannot understand why No 4 and No18 are blanked in the 2nd right column although they have 3 BM foci.

A9.I couldn't understand this comment, but I tried to make all the corrections you suggested in Table 1.

#Table 2

Q10--This is similar to the above about Table 1. The ‘Primary Diagnosis’ or ‘Histopathological Subtype’ should be re-considered. For example, why ‘Lung Cancer’ is ‘Primary Diagnosis’ and ‘Adenocarcinoma’ is ‘Histopathological Subtype’? Probably, ‘Lung Carcinoma’ is ‘Primary Diagnosis’, then comes ‘Adenocarcinoma’ as the ‘Histopathological Subtype’. Or, just ‘Lung adenocarcinoma’ would suffice as ‘Primary Diagnosis’. Anyway, this table should be accordingly revised by the help of pathologists.

-If this table intends to show ‘Staining Score’, the materials and methos should sufficiently address this, which I will tell later.

A10.Again, based on your valuable advice, the incorrect definitions in Table 2 were tried to be written correctly and highlighted in yellow.

#Figure 7 and 8

Q11.The IHC photos should be shown with much higher magnification (x200 or more). With such a low mag., no one can recognize the brown pigment on endothelium within the tumor tissues.

A11.This study was conducted with a limited budget, and although your advice is very valuable, figures 7 and 8 have been removed from the article, regretfully, as imaging with a new higher magnification is not possible.

Q12#page 3, line 132-133

-anti-PSMA antibody: The lot number, the product ID, and/or whether they are polyclonal or monoclonal should be clearly shown. The full information of this antibody is a must.

-treated with for 1 hour and yet incubated for 20 minutes??? After all, this means that ‘incubated with the antibody for 1 hour. 20 min.’???

A12.New description added to line 158-170 as advised.

-Nex >>> Next

Q13-The quantification strategy is quite unclear. What is staining score??? Why can staining score of 3 or less be low, and that of higher than 3 be high???

-After all, this section ‘IHC staining with PSMA antigen’ is extremely poorly described.

A13.New description added to line 165-170 as advised.

Q14#Figure 1-8

I am afraid ‘red arrows’, ‘dashed red arrows’, or so are not colored rightly, or they do not indicate the right lesion. For example, ‘blue arrows’ in Figure 1 look red. Is this due to the conversion of digital files? Anyway, please check the thoroughness of the legends all over.

A14. All figures, arrows and explanations were reviewed and necessary corrections were made.

Q15#page 4, line 151, ‘NET’: What is this NET? Is this grade 1, 2, or 3? This should be NET grade 3, I guess. A more correct, pinpointed term should be applied.

A15.Changed to NET 3 upon your recommendation

Q16.#Some minor wording/grammatical errors I have noticed. Please check the MS again.

A16.The entire text has been reviewed for spelling errors as per your recommendation and attempts have been made to correct any deficiencies.

Q17.-page 2, line 67-68: (18F-FDG), positron…(PET/CT) which is >>> (18F-FDG) positron…(PET/CT), which is

A17.Similarity and standardization in abbreviation was achieved.

Q18-page 3, line 121: F-18 FDG >>> 18F-FDG (?)

A18.Standardization of the 18F-FDG PET/CT terminology was achieved and necessary corrections were made.

Q19.-Figure legend of figure 2: 50%.  .2cm axial diameter >>> 50%. 2cm axial diameter

A19.Figure 2 and its legend were deleted at the suggestion of another reviewer.

Round 2

Reviewer 3 Report

Comments and Suggestions for Authors

This paper has been fairly revised according to many comments of 3 reviewers. But if the paper is really intended for publication, I have to raise some more comments about it.

#Simple Summary, line 26-27, probably this sentence is not grammatically correct: had PSMA-positive with >>> had PSMA-positivity with, with but only >>> with only, one patient had PSMA-negative >>> one patient having PSMA-negative. (The similar point is applicable to line 181)

#Conclusion, line 62-63, ‘While the results…are clearly needed’: I guess this sentence was inserted into the conclusion according to the comments of reviewers. But this kind of sentence stating ‘limitations’ should be placed at the very end of it if it is inserted. There is no need to weaken the content of a study at the very beginning of it. Furthermore, the flow of discussion/concept in this ‘Conclusion’ is not fluent or eloquent. It comes this way, and then, goes that way, in kind of a haphazard fashion. It reads rather strange.

#This is pertinent to the above. I can see what the authors try to say in ‘Conclusion’ in line 314 to 332’, and the authors’ endeavor to incorporate the reviewers’ comments into it, but because of the mixture of the authors as well as the viewers in it, it is rather clumsy and hard to understand in the right way, I guess. The comments of the authors are likely true, but more consolidated and straightforward way of description is probably warranted.

#The section ‘IHC Staining with PSMA Antigen’

-Still the information of the PSMA is insufficient. What is the dilution? What is the producer (company)?

-line 159, ‘fixed’ in xylene’: It is not that the sections are ‘fixed’ in xylene. To treat them in xylene is the process for rehydrating them for staining. To use ‘fixed’ here is not correct.

-Still the strategy for quantification of PSMA-stained slides is unclear. The four-tiered system (0-3), which probably expresses the ‘percentage’, seems okay, but where is the intensity??? Why is it possible to multiply the staining percentage by the staining intensity, with the latter noted nowhere??? Regarding this, the ‘Table 2’ shows like, for instance, ‘1X1:1’. The second ‘1’ in ‘1X1’ happens to be the intensity???

#Table 1

ID: Is this ‘intraductal’? If so, this should be clearly stated in the caption of this table.

Comments on the Quality of English Language

Minor corrections may be necessary. 

Author Response

Q1.#Simple Summary, line 26-27, probably this sentence is not grammatically correct: had PSMA-positive with >>> had PSMA-positivity with, with but only >>> with only, one patient had PSMA-negative >>> one patient having PSMA-negative. (The similar point is applicable to line 181)

 A1. ‘’68Ga-PSMA-11 PET/CT revealed PSMA-positive brain metastases in 26 of 27 patients; only a single patient exhibited PSMA-negative lesion.’’ On line 26 and line 194, an attempt was made to correct the sentence in the most grammatical way possible upon your recommendation.

Q2.#Conclusion, line 62-63, ‘While the results…are clearly needed’: I guess this sentence was inserted into the conclusion according to the comments of reviewers. But this kind of sentence stating ‘limitations’ should be placed at the very end of it if it is inserted. There is no need to weaken the content of a study at the very beginning of it. Furthermore, the flow of discussion/concept in this ‘Conclusion’ is not fluent or eloquent. It comes this way, and then, goes that way, in kind of a haphazard fashion. It reads rather strange.

  A2. Based on your recommendation, the first sentence on line 62 was removed from the conclusion and the conclusion part was tried to be made more fluent.

Q3.##This is pertinent to the above. I can see what the authors try to say in ‘Conclusion’ in line 314 to 332’, and the authors’ endeavor to incorporate the reviewers’ comments into it, but because of the mixture of the authors as well as the viewers in it, it is rather clumsy and hard to understand in the right way, I guess. The comments of the authors are likely true, but more consolidated and straightforward way of description is probably warranted.

  A3. I tried to make the narration as academic and fluent as possible, upon your recommendation.

Q4.#The section ‘IHC Staining with PSMA Antigen’ Still the information of the PSMA is insufficient. What is the dilution?

  A4. Dilution range: 1:20. Added line 164

Q5.#- What is the producer (company)?

  A5. PSMA (EP192) Rabbit Monoclonal Primary Antibody (Roche / Cell Marque) (EP192) Added to material method section as advised. Line 157.

Q6.#--line 159, ‘fixed’ in xylene’: It is not that the sections are ‘fixed’ in xylene. To treat them in xylene is the process for rehydrating them for staining. To use ‘fixed’ here is not correct.

  A6. We are so sorry.  Section are fixed formaline. Line 158.

Q7.-Still the strategy for quantification of PSMA-stained slides is unclear. The four-tiered system (0-3), which probably expresses the ‘percentage’, seems okay, but where is the intensity??? Why is it possible to multiply the staining percentage by the staining intensity, with the latter noted nowhere??? Regarding this, the ‘Table 2’ shows like, for instance, ‘1X1:1’. The second ‘1’ in ‘1X1’ happens to be the intensity???

 A7. Table 2  has been revised and clarified based on your recommendations and explanation added to Line 232

Q8.-#Table 1

ID: Is this ‘intraductal’? If so, this should be clearly stated in the caption of this table.

A8. *ID: Invasive Ductal abbreviation explanation added to the bottom of the table

Best regards
